# Deep Learning Based Prediction of Gas Chromatographic Retention Indices for a Wide Variety of Polar and Mid-Polar Liquid Stationary Phases

**DOI:** 10.3390/ijms22179194

**Published:** 2021-08-25

**Authors:** Dmitriy D. Matyushin, Anastasia Yu. Sholokhova, Aleksey K. Buryak

**Affiliations:** A.N. Frumkin Institute of Physical Chemistry and Electrochemistry, Russian Academy of Sciences, 31 Leninsky Prospect, GSP-1, 119071 Moscow, Russia; dm.matiushin@mail.ru (D.D.M.); akburyak@mail.ru (A.K.B.)

**Keywords:** gas chromatography, deep learning, untargeted analysis, retention index, QSPR, QSRR

## Abstract

Prediction of gas chromatographic retention indices based on compound structure is an important task for analytical chemistry. The predicted retention indices can be used as a reference in a mass spectrometry library search despite the fact that their accuracy is worse in comparison with the experimental reference ones. In the last few years, deep learning was applied for this task. The use of deep learning drastically improved the accuracy of retention index prediction for non-polar stationary phases. In this work, we demonstrate for the first time the use of deep learning for retention index prediction on polar (e.g., polyethylene glycol, DB-WAX) and mid-polar (e.g., DB-624, DB-210, DB-1701, OV-17) stationary phases. The achieved accuracy lies in the range of 16–50 in terms of the mean absolute error for several stationary phases and test data sets. We also demonstrate that our approach can be directly applied to the prediction of the second dimension retention times (GC × GC) if a large enough data set is available. The achieved accuracy is considerably better compared with the previous results obtained using linear quantitative structure-retention relationships and ACD ChromGenius software. The source code and pre-trained models are available online.

## 1. Introduction

Gas chromatographic retention index (RI) is a value that does not strongly depend on particular chromatographic conditions and characterizes the ability of a given stationary phase (SP) to retain a given molecule [1]. RI is transferable between different gas chromatographic conditions and instrument setups. Despite the fact that there is a minor dependence of RI on temperature and other factors [2], RI can be considered as mainly dependent on SP and the structure of the molecule [2]. RI is widely used in gas chromatography (GC), in particular in metabolomics [3], in the analysis of essential oils [4], plant volatiles [5], flavors and fragrances [5], and in the environmental analysis [6]. RI can be used [7,8] as an additional constraint for the library search in gas chromatography-mass spectrometry (GC-MS).

All available reference databases contain RI for less than 150,000 compounds. Mass spectra are available for 2–3 times greater number of compounds. At the same time, millions of compounds are known and described. Methods of mass spectra prediction and analysis for GC-MS identification without reference databases are available [9,10]. Predicted RI can be used as a reference for the GC-MS library search [7,11,12,13], in particular in metabolomic applications. Hence, the RI prediction is an important task.

There are multiple previous works devoted to the RI prediction, which is also referred to as quantitative structure-retention relationships (QSRR). The majority of such works consider very small data sets containing several dozen compounds that are not chemically diverse. In these cases, quite suitable accuracy is often obtained for very small test sets. However, such models are not versatile, and usually, the limits of applicability are not known. Earlier QSRR works devoted to narrow data sets are reviewed in references [14,15,16]. An important task is the development of versatile RI prediction models that are applicable to almost arbitrary structures. There are several works (e.g., the most recent works [17,18,19]) that are devoted to RI prediction for diverse compounds and use data sets ranging in size from hundreds to tens of thousands of compounds. Most of such works, except for the most recent ones, are extensively reviewed in our previous work [17].

Complex machine learning models based on deep neural networks of complex architecture (including convolutional, recurrent, and residual neural networks) that are able to extract information directly from raw features rather than use human-engineered features are usually called the term “deep learning”. Deep learning is one of the most rapidly developing methodologies in all fields of science [20]. Deep learning models are often much more accurate than models based on hard-coded features when large enough training sets are available. Over the last two years, at least four works [17,18,19,21] applied deep learning methods for RI prediction using large (tens of thousands compounds) data sets. Another work [22] uses deep learning for the prediction of GC retention time of some steroids. In all these cases, deep learning methods predict much better than common QSRR methods. Common QSRR methods provide mean and median errors of RI prediction for non-polar SP (e.g., DB-1, DB-5, squalane, OV-101) in the range of 40–60 and 30–50, respectively. For example, a popular functional group contributions-based model [23] gives a median absolute error of 46. QSRR models [24] trained using the Flavornet database (https://www.flavornet.org/, accessed on 22 August 2021) provide mean and median errors of about 50 [17] for non-polar SP. For deep learning models, mean and median errors below 30 and 20, respectively, are achievable for the same or similar data sets [17,18,19].

Almost all “versatile” RI prediction models [17] are developed for non-polar SP. There are several such works for polar SP (e.g., polyethylene glycol (PEG), DB-WAX). The NIST functional group contributions-based model is trained for polar SP [23] (median error 65). QSRR models for RI of essential oils, flavors, and fragrances on polar SP are developed [25,26] using data sets from references [4,27]. Most other works consider very small (less than 100 compounds) narrow data sets, and these models cannot be considered as versatile.

Several works are devoted to accurate RI prediction for mid-polar SP. In reference [11], an accurate retention prediction model for DB-624 SP (6% cyanopropylphenyl 94% dimethyl polysiloxane) was developed and used to improve the GC-MS library search. A data set with more than 500 compounds is published. In reference [24], OV-17 SP (50% diphenyl 50% dimethyl siloxane) is also considered along with polar and non-polar SP. In reference [28], an accurate QSRR model for second dimension retention times and RI in two-dimensional GC was built. BPX50 (50% phenyl polysilphenylene siloxane) is used as a second dimension SP. A data set with more than 800 compounds is published. The above-mentioned works [11,28] use proprietary ACD ChromGenius software for building an accurate QSRR model, and there are only sparse explanations of how exactly retention is predicted. Some works consider relatively large data sets consisting of compounds of very similar structures, e.g., polychlorinated biphenyls [29] and polybrominated diphenyl ethers [30]. Neural networks were used in these works. Such models are specialized for a given class of compounds and cannot be directly generalized for diverse compounds.

RI prediction for polar (e.g., DB-WAX) and mid-polar (e.g., DB-624, DB-210, DB-1701, OV-17) SP is an important task. For polar SP, reference experimental RI is available for approximately 10,000 compounds, which is several times less than for non-polar SP. For mid-polar SP, large enough RI databases are absent. However, mid-polar SP are widely used in analytical practice [31,32,33,34], and RI prediction for these SP can be used for the GC-MS analysis of volatiles [11]. RI prediction for a wide variety of SP can be used for SP selection for given analytes and for the planning of an experiment. For polar SP, the same deep learning techniques can also be applied as used for non-polar SP. Several thousand compounds are enough for such methods to achieve reasonable accuracy. For mid-polar SP, the small size of the data set makes prediction more complicated, and this task probably requires new approaches.

The aims of this work are the application of previously developed (for non-polar stationary phases) methods to retention index prediction on polar stationary phases, development of accurate and versatile retention index prediction models for mid-polar stationary phases (e.g., DB-624), and testing whether these approaches are usable for prediction of second dimension retention times. Only free and open-source software is used; source code and pre-trained models are published online: https://doi.org/10.6084/m9.figshare.14602317, accessed on 22 August 2021.

## 2. Results

### 2.1. Data Sets and Machine Learning Models

For the training of neural networks, the NIST 17 database was used. Two more external test sets with data for polar SP were used (see Table 1). All data sets, except for NIST 17, used in this work are listed in Table 1 with the designations used below, data sources, and information about the SP type. Some compounds were removed from these data sets according to the criteria given in the Methods section (Section 3.3). n-⁠Alkanes were also removed from these data sets. Appendix A, contains information about the types of polar SP, data set sizes, and overlaps between data sets used in this work. Appendix A, shows the RI distributions for these data sets. An extensive description of data sets processing and training-test split is also given in the Methods section (Section 3.3).

The machine learning and deep learning models used in this work are schematically depicted in Figure 1. CNN and MLP are convolutional neural network and multilayer perceptron, respectively, trained using data for standard and semi-standard non-polar SP. The structure and hyperparameters of these neural networks are very close to the models used in work [17]. Two other deep learning models, referred to as CNNPolar and MLPPolar, are the same as CNN and MLP but are trained with RI for standard polar SP. For BPX50, OV-17, and DB-624 stationary phases, we use a second-level machine learning model (see Figure 1C). The key idea is to use RI values for non-polar and polar SP as input features (e.g., molecular descriptors) for RI prediction for mid-polar SP. Support vector regression (SVR) was used for this purpose. The detailed description of all machine learning models used in this work, values of hyperparameters, and implementation notes are given in the Methods section (Section 3.1, Section 3.2, and Section 3.4).

### 2.2. Polar Stationary Phases

The mean absolute error (MAE), mean percentage error (MPE), root mean square error (RMSE), median absolute error (MdAE), and median percentage error (MdPE) are used as accuracy measures in this work. Table 2 shows the accuracy of the considered deep learning models for RI prediction for polar SP. The results for CNNPolar, MLPPolar, and for the simple average of their predictions are demonstrated. The accuracy of previously reported models is also shown. The accuracy for NIST 17 is the 5-fold cross-validation accuracy (mean over five subsets used as test sets). For FLAVORS and ESSOILS, the models were retrained with the corresponding test compounds removed from the training set. For FLAVORS and ESSOILS, the results from reference [26] and reference [25] are given, respectively.

The values that are given as previous results for the NIST data set were obtained for the much smaller NIST 05 data set using the functional group contributions method [23]. To the best of our knowledge, this is the best RI prediction method for polar SP that was tested on large and diverse data sets. The results that are given in that work [23] are calculated using a compounds-based data set, i.e., the median of all RI values for each compound is used (one data entry for one compound). We use all data entries from the NIST 17 database without such grouping. On average, there are more than nine data entries per compound. It is not clear if the accuracies computed in such different ways and for different versions of NIST are comparable. Therefore, we trained the linear functional group contributions model [23] using our data set (no training-test split was used because the linear model is robust to overfitting) and also presented its accuracy in Table 2.

Table 2 clearly shows that our model significantly outperforms the previously described linear QSRR models. This is expected, taking into account the suitable accuracy of these models for the prediction of RI on non-polar SP [17]. However, for polar SP, the advantage of deep learning models is even greater. The MLPPolar and CNNPolar models have similar accuracy, and simple averaging of their results allows further considerable improvement of the accuracy, as shown in Table 2. Figure 2 shows the predicted versus reference values for the average outputs of CNNPolar and MLPPolar, error distributions for deep learning models, and accuracies for five cross-validation subsets. The data in Figure 2 correspond to the NIST 17 database (polar SP). Similar data for ESSOILS and FLAVORS data sets are also shown in Figure 3A–C.

For the average of the outputs of CNNPolar and MLPPolar (the NIST data set was used), the deviation between the predicted and reference RI values for 90% of the data records is less than 93 RI units. For the linear functional group contributions-based model [23], this range is considerably wider: 90% of data records fall into the range of ±104 units compared with the reference values. This value corresponds to the linear functional group contributions-based model trained using our data set. The coefficients of determination R^2^ between the predicted and observed values are 0.966 and 0.913 for the average of the outputs of CNNPolar and MLPPolar and for the linear functional group contributions-based model, respectively.

### 2.3. DB-624 and OV-17 Data Sets

RI prediction for mid-polar SP is much more challenging compared with polar and non-polar SP. The main reason is the lack of large enough (thousands of compounds) data sets. In reference [11], suitable prediction accuracy was achieved for the DB-624 data set using proprietary ChromGenius software. We tried to train our neural networks from scratch for this data set and did not achieve suitable performance. When we used various variants of transfer learning (i.e., reusing the parameters of neural networks obtained for other SP), we achieved accuracy similar to that given in reference [11] for the ChromGenius model: MAE = ~30 and MPE = ~3%. However, the employment of the second-level model allows us to achieve significantly better accuracy (see Table 3). In addition to accuracy improvement, the use of the second-level model is much less resource-demanding at the training stage compared with neural networks/transfer learning, and also, the SVR model is more robust to overfitting. The use of random forest and gradient boosting instead of SVR is also possible; however, these methods do not improve accuracy but are prone to overfitting (especially gradient boosting) and result in much larger model files. Model development and hyperparameters tuning were made for the DB-624 data set. All accuracy measurements for the OV-17 and other data sets described below were performed after the final model architecture and hyperparameters were selected.

Table 3 shows the accuracy of the second-level models for DB-624 and OV-17 data sets. For DB-624, we used the same training-test split as in reference [11]. The accuracies are given for the test set. However, we removed 20 compounds from the training set that are stereoisomers or isotopomers of the compounds contained in the test set, so our approach is even stricter. For OV-17, the 10-fold cross-validation accuracies are given for the whole data set.

Table 3 shows that our model significantly outperforms those previously published. For DB-624, we also performed cross-validation using the training set described in reference [11]. In this case, we obtained similar accuracy (MAE = 24.9; MdAE = 11.8; MPE = 2.47%; MdPE = 1.29%). In reference [11], there are two models for the DB-624 data set: the ChromGenius model Icg and the RapidMiner model Irm. The data for the ChromGenius model are given in Table 3. If we consider the composite model Icomp = 0.84 × Icg + 0.16 × Irm, we slightly improve the accuracy compared with Icg: MAE = 31.2; MdAE = 17.8; MPE = 2.98%; MdPE = 2.08%. However, it is still considerably worse compared with our results.

For 90% of compounds, the prediction of our model deviates from the correct value for less than 60 units (72 units for the ChromGenius model from reference [11]). The coefficients of determination R^2^ between the predicted and observed values are 0.989 and 0.976 for our model and the ChromGenius model, respectively. These additional accuracy measures also indicate that our model is considerably more accurate than the previously developed.

Therefore, it can be concluded that our second-level model is more accurate compared with state-of-the-art QSRR models for mid-polar SP. Figure 3 demonstrates error distributions, predicted versus reference values, and the comparison of the results of this work with the previous results for the FLAVORS, ESSOILS, DB-624, and OV-17 data sets. Finally, it should be noted that, unlike the authors of reference [11], we do not use closed-source software in our work, and the full RI prediction pipeline from descriptor computation to final prediction is transparent Java code.

### 2.4. Prediction of Second Dimension Retention Times

RI for 1D GC cannot be directly applied for retention in the second dimension. There is a smooth dependence of the second dimension retention time on the difference between one-dimensional RI for SP used in the first and in the second dimensions [31]. Such an approach can probably be used for second dimension retention prediction, while we can accurately predict RI for a wide variety of SP. However, the relative accuracy of such difference will be much worse compared with the relative accuracies of one-dimensional RI for each of SP because the value will decrease after subtraction, and the errors will be added.

In reference [28], a relatively large data set for second dimension retention times is provided. The authors of that work use ACD ChromGenius to develop a QSRR model for second dimension retention time prediction. They predict both second dimension retention times and polyethyleneglycol-based second dimension RI. These values are linearly interconnected, so in order to compare our model with the tools used by the authors of reference [28], we can use any of them. We used second dimension retention times; however, all the conclusions given below are applicable to polyethyleneglycol-based second dimension RI.

We trained our second-level model to predict second dimension retention times (in milliseconds) using the same training set as used by the authors of that work [28]. We then evaluated our model using two data sets that were used by the authors of reference [28] for evaluation of their model. The results of the comparison are given in Table 4. Our model performs better than ACD ChromGenius and can be used for the creation of QSRR models for second dimension retention times and indices if a large enough data set is available for training (at least ~150 compounds).

### 2.5. Further Testing and Applications

For further external testing of our models for DB-624 SP, we considered one more data set: SEDB624. We excluded from the training set for the second-level model the compounds that are also contained in this data set and retrained the second-level model. The accuracy for SEDB624 is given in Table 5. A plot with predicted versus reference values for this data set is shown in Figure 4. High accuracy is observed.

RI prediction for other mid-polar SP, such as DB-210 (50% trifluoropropylmethyl 50% dimethyl polysiloxane) and DB-1701 (14% cyanopropylphenyl 86% dimethyl polysiloxane), is even more complicated due to the absence of large enough data sets. We used a data set with 36 compounds for DB-1701 and a data set with 130 compounds for DB-210. The latter data set is not that small, but it consists of several series of homologues. Despite the fact that it contains compounds of different classes, it cannot be considered as really diverse. These data sets can be used for training the SVR second-level model, but it is not possible to conclude that this model will work well for other compounds and to determine the limits of applicability. Even if such a complex model (with hundreds of input features) for such small data sets would not overfit for particular compounds (it can be checked with cross-validation), it is not possible to be sure that it will not “overfit” for particular classes of compounds that constitute the data set, and that the model has suitable accuracy for anything else.

In the case of DB-624 and OV-17, we use predictions for those SP for which we can predict well as the inputs for the models for those SP for which large data sets are not available. For DB-210 and DB-1701, we decided to use the same idea but keep the model as simple as possible with very few fitted parameters. The GC retention can be approximately represented [35] as the sum of the products of analyte-specific and SP-specific parameters:log*k* = c + eE + sS + aA + bB + lL,(1)
where *k*—the retention factor (isothermal GC); lowercase letters denote SP-specific parameters, and uppercase letters denote analyte-specific parameters. If this equation is true, it has the following consequence: log*k* for an arbitrary liquid SP can be approximated as a linear combination of log*k* for not more than five other SP. For isothermal GC, logk is nearly linearly linked with RI (see Appendix A). Using this idea, for RI on DB-210 and on DB-1701, we developed the following equations:RI_DB-1701_ = 0.949 × RI_DB-624_ + 0.0769 × RI_DB-WAX_(2)
RI_DB-210_ = −3.3267 × RI_Squalane_ − 0.3007 × RI_DB-1_ + 3.0737 × RI_DB-5_ + 1.72 × RI_DB-624_ − 0.1071 × RI_DB-WAX_(3)

All RI values on the right side are predicted using the methods described above. To develop these equations for each SP, at the first stage, we selected that SP, the RI values for which are the most correlated with the RI values for the target SP, i.e., we selected the best such equation with only one SP on the right side. For DB-1701, it was DB-624, and for DB-210, it was DB-WAX. Then, we step by step added more and more SP to the right side. Each time we added the SP that gives the most accuracy gain. We continued the process until adding a new term to the right side of the equation stopped giving a considerable gain (at least ~3–5 units in terms of MAE after adding the next term). As candidates for use on the right side, we considered only common SP for which we had a large number of training compounds, and we avoided trying SP of very similar polarity together.

RI for DB-1701 are strongly correlated with RI for DB-624. These SP have similar selectivity, but DB-1701 is more polar. So, the proposed equation has a clear physical meaning. It should also be noted that DB-1701 is the smallest data set (36 compounds), and the simplest equation is preferred in this case to avoid any form of overfitting.

The accuracy of retention prediction using these equations is given in Table 5. Fairly suitable accuracy was obtained for these SP, especially for DB-1701. Such an approach opens up the opportunity to predict the RI of almost any compound with reasonable accuracy for an almost arbitrary liquid SP, as long as at least a small data set is available. The plots with predicted versus reference values for these data sets are shown in Figure 4.

## 3. Methods

### 3.1. Deep Neural Networks

The deep learning models, referred to as CNN and MLP, are equal to the CNN1D and MLP models, respectively, from our previous work [17]. Input features, architecture, early stopping criteria, and all hyperparameters, except for the batch size, are the same as used in that work [17]. Batch size 256 was used instead of 16 in order to accelerate training. A total of 50,000 training iterations were performed, and the neural network parameters for the iteration with the best validation set score were used later to generate input for the second-level model. The same data set as in the previous work [17] was used for the training of these models (RI for non-polar SP).

Weights and biases from MLP and CNN after 50,000 training iterations were used as the initial values of the neural network parameters for MLPPolar and CNNPolar. It can also be considered as transfer learning. A total of 10,000 training iterations were performed in this case, and the neural network parameters for the iteration with the best validation set score were used later. All other hyperparameters were the same as for MLP and CNN. For all neural networks, 10% of the compounds from the training set were isolated before training and used for validation and early stopping.

Neural networks use information about SP together with representations of the molecule structure. MLP and CNN use the same SP encoding as used in the previous work [17], and CNNPolar and MLPPolar use the simple one-hot encoding of SP types. Twenty of the most abundant in the NIST 17 database standard polar SP were considered separately, and the rest were grouped into the single SP type “Other polar”. The SP types are listed in the Appendix A. We strictly follow the names and types of SP given in the NIST 17 database. In fact, some of these SP names are different names of the same SP, and others are ambiguous and can correspond to different SP. In order to keep the structure of CNNPolar and MLPPolar the same as the structure of CNN and MLP, we use an SP encoding input array with a length of 38 for all neural networks. However, only 21 of these features are meaningful for CNNPolar and MLPPolar, and others are fixed and not used. CNNPolar and MLPPolar are used to predict RI for polar SP (e.g., DB-WAX, Carbowax 20 M, etc.). For mid-polar SP, second-level models are used.

### 3.2. Second-Level Models for Mid-Polar Stationary Phases

SVR with exponential kernel (σ = 31.0) is used with the following hyperparameters: threshold parameter ε = 0.82, soft margin penalty parameter C = 31·10^3^, tolerance T = 2.7. The target RI values (labels) were divided by 1000 at the training stage, and consequently, the SVR prediction is multiplied by 1000 in order to obtain the predicted RI value.

The SVR model was trained with the following input features: RI for all supported non-polar and polar SP types predicted with CNN, MLP, CNNPolar, and MLPPolar neural networks, and molecular descriptors. For each of the CNN and MLP neural networks, 36 values for various types of non-polar SP were obtained for each compound. For each of the CNNPolar and MLPPolar neural networks, 21 values for various types of polar SP were obtained for each compound. All RI values were divided by 1000 in order to avoid too large input features. We concatenated these 114 features with 327 molecular descriptors. Finally, a set of 441 input features was used for the training of SVR. The molecular descriptors set consists of 243 molecular descriptors calculated with Chemistry Development Kit (CDK), version 2.3 [36], and 84 functional group counters. The set of descriptors is exactly equal to the set used in our previous work [17] and is described there. Scaling factors for descriptors were not recalculated during training and were taken from previously published models [37].

### 3.3. Data Sets Preprocessing and Training-Test Split

Compounds that were not supported by our software [37] or that could not be extracted from the database were removed. Exclusion criteria (e.g., uncommon elements, such as selenium, too large structures) are given in our previous work [17]. For non-polar SP, the same data set was used as described in the previous work [17]. For polar SP, the corresponding data entries were extracted in the same fashion as for non-polar SP. The resulting data set contained 89,086 data entries and 9408 compounds. Multiple data entries are often given for one compound in NIST 17.

When we evaluated our models for polar SP for each of the test sets, all compounds that were contained in the test set were excluded from both training sets consisting of data for polar and non-polar SP. The neural networks were then trained from scratch using the obtained training sets for training and early stopping. Consequently, the compounds from the test set were not seen by our models at all stages of training and were not even used for early stopping. Exclusions are compounds-based, i.e., we exclude all data entries that correspond to the excluded compound. Stereoisomers (*cis-trans* and optical) are treated as equal compounds following the previous work [17]. The data set consisting of data for polar SP extracted from NIST 17 was split into five subsets, and each of them was used as a test set (5-fold cross-validation).

For training of the second-level model for mid-polar SP, we require pre-trained neural networks for polar and non-polar SP. For the training of these neural networks, we excluded all compounds that were contained in all non-NIST data sets considered in this work (see Table 1) from the NIST database. The obtained neural networks were used for all studies considering mid-polar SP (including GC × GC). In these cases, the compounds used to measure accuracy were also not used for training at any of the stages.

For the previously described in reference [11,24,25,26] models (for DB-624, FLAVORS, and ESSOILS data sets), we calculated accuracy measures on our own, using the observed and predicted values published alongside the corresponding works.

### 3.4. Implementation and Software

This work is based on our previous work [17] and the corresponding software published online [37]. In this work, we added three new Java classes to that software; however, the rest of the code was not changed. The Deeplearning4j framework (https://deeplearning4j.org/, accessed on 22 August 2021), version 1.0.0-beta6, was used for deep learning, CDK 2.3 was used for chemoinformatics tasks. The Smile (http://haifengl.github.io/, accessed on 22 August 2021) framework, version 2.5.3, was used for SVR. All dependencies and software are free and open-source. The source code, pre-trained model parameters, source code of simple graphical user interface, and corresponding instructions are provided in the online repository: https://doi.org/10.6084/m9.figshare.14602317, accessed on 22 August 2021.

## 4. Conclusions

Deep learning models that were previously developed to predict retention indices for non-polar stationary phases also work enormously well for polar stationary phases. The mean and median errors are 1.5–2 times less than those achieved in previous works. For mid-polar stationary phases, when large enough training sets (~150 compounds) are available, a second-level model can be trained for accurate retention prediction for both single-dimensional GC and GC × GC. This approach significantly outperforms simple quantitative structure-retention relationships and even outperforms ACD ChromGenius software in this task. In cases when large data sets are not available, simpler linear models are proposed, and their accuracy is still suitable. We use the retention index values for those stationary phases for which large training sets are available as input features for models that predict the retention indices for other stationary phases. In all cases, the mean absolute error is not more than ~50 units for the retention index, and this approach can be applied to an almost arbitrary liquid stationary phase.

## Figures and Tables

**Figure 1 ijms-22-09194-f001:**
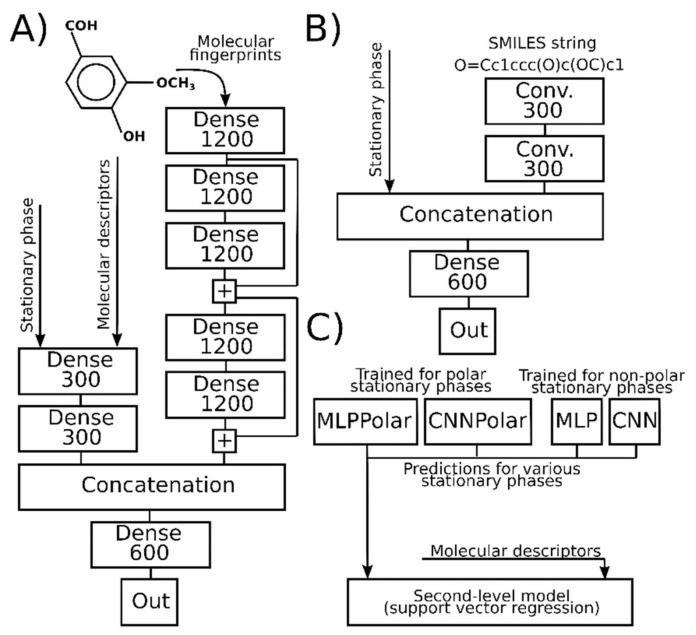
The machine learning models that were used in this work. Dense—a fully connected layer. Conv.—a convolutional layer. “+” indicates element-wise additions. The number in the block denoting the layer indicates the number of output nodes/channels. (**A**)—the depiction of MLP and MLPPolar models. (**B**)—the depiction of CNN and CNNPolar models. (**C**)—the second-level support vector regression model.

**Figure 2 ijms-22-09194-f002:**
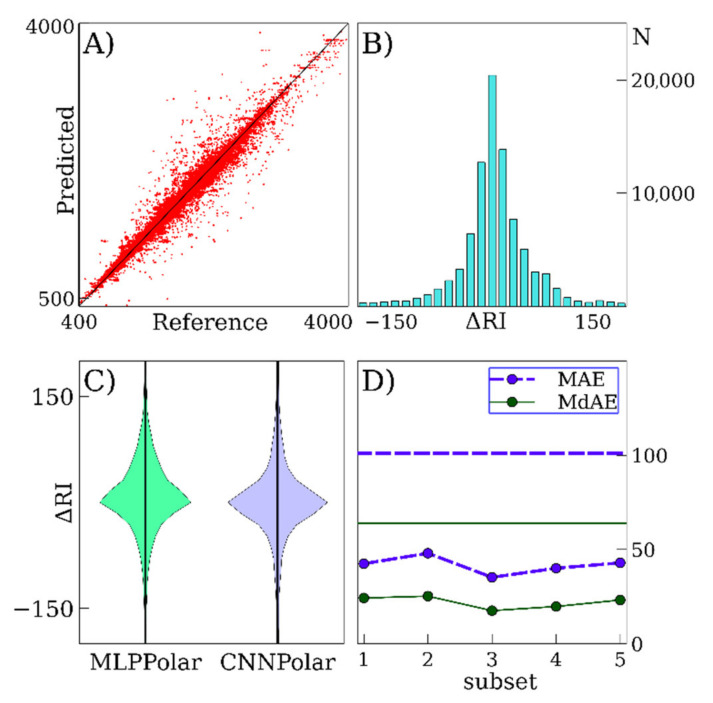
Retention index prediction for polar stationary phases using the NIST 17 database for training and validation. The average of CNNPolar and MLPPolar predictions is considered unless otherwise specified. (**A**)—predicted versus reference values for the full database (cross-validation). (**B**)—distributions of errors. (**C**)—distributions of errors for different deep learning models. (**D**)—mean and median absolute errors of prediction for the cross-validation subsets of NIST 17. The horizontal lines denote the results for NIST 05 given in reference [23].

**Figure 3 ijms-22-09194-f003:**
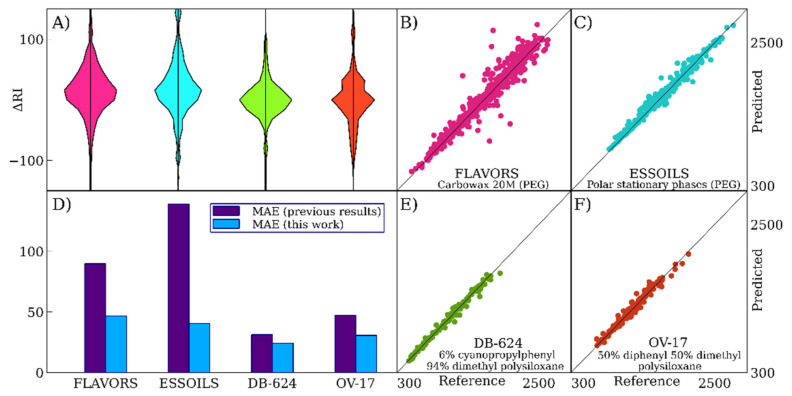
The accuracy of retention index prediction for various data sets and stationary phases. (**A**)—distributions of errors for various data sets. (**B**,**C**)—predicted versus reference values for FLAVORS and ESSOILS data sets (polar stationary phases). (**D**)—a comparison with the previously published results [11,24,25,26]. (**E**,**F**)—predicted versus reference values for DB-624 and OV-17 data sets (mid-polar stationary phases).

**Figure 4 ijms-22-09194-f004:**
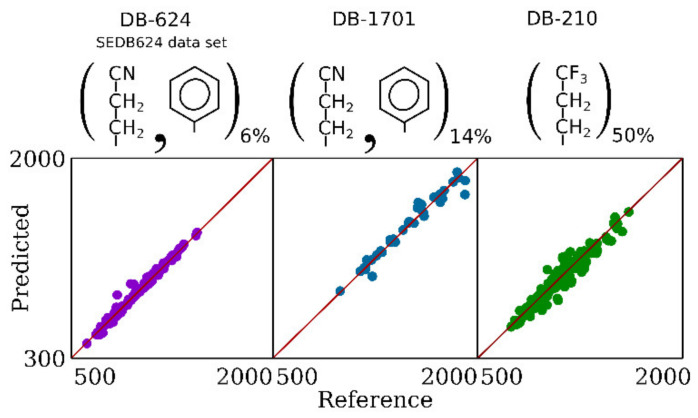
Comparison of predicted and reference values for SEDB624, DB-1701, and DB-210 data sets.

**Table 1 ijms-22-09194-t001:** The data sets considered in this work (except for the NIST 17 database).

Designation	N	Compounds	Stationary Phase	Reference ^a^
FLAVORS	1169	Flavors and fragrances	Carbowax 20 M	[26,27]
ESSOILS	427	Essential oils components	Various polar SP	[4,25]
DB-624	545 ^b^	Various aliphatic and aromatic alcohols, esters, ethers, aldehydes, sulfur-containing compounds, heterocycles, nitriles and other compounds	DB-624	[11]
OV-17	192	Odorants (the Flavornet database https://www.flavornet.org/, accessed on 22 August 2021)	OV-17	[24]
BPX50_2D	859 ^c^	The diverse set of environmental-related compounds: pesticides, organophosphates, esters, polyaromatic compounds, polychlorinated biphenyl congeners, polychlorinated dioxins, bisphenols, etc.	BPX50	[28]
SEDB624	130	Series of homologues of ketones, aldehydes, alcohols, alkylbenzenes, alkenes, chloroalkenes, cycloalkenes, esters, and other compounds	DB-624	[31]
DB-1701	36	Flavors and fragrances	DB-1701	[32]
DB-210	130	The same compounds as in the SEDB624 data set	DB-210	[31]

^a^—both the original source and the actually used secondary source are specified, when applicable. The number N of data records actually used in this work is given. ^b^—the data set is split into the training and test sets in the original work (396 and 149 data entries, respectively). ^c^—the data set is split into the training, test, and external test sets in the original work (359, 168, and 332 data entries, respectively).

**Table 2 ijms-22-09194-t002:** The accuracy of retention index prediction for polar stationary phases.

Model	Metric	Data Sets		
		NIST	FLAVORS	ESSOILS
CNNPolar	RMSE	92.0	92.3	70.1
	MAE	47.3	53.8	46.9
	MdAE	23.4	32.9	30.8
	MPE	3.12	3.40	2.84
	MdPE	1.56	2.21	1.73
MLPPolar	RMSE	82.5	94.2	64.6
	MAE	45.8	52.3	45.6
	MdAE	27.4	27.5	31.0
	MPE	3.07	3.26	2.77
	MdPE	1.80	1.86	1.89
Average	RMSE	80.3	86.1	58.8
	MAE	41.7	46.6	40.4
	MdAE	22.0	26.1	26.4
	MPE	2.77	2.93	2.47
	MdPE	1.45	1.74	1.59
Previous works [23,25,26]	RMSE	154	125.4	177.3
	MAE	101	89.8	139.0
	MdAE	64	68.6	123.5
	MPE	5.7	5.76	8.42
	MdPE	3.9	4.49	7.09
Linear model [23]	RMSE	129.1	157.6	91.8
	MAE	72.5	92.4	61.7
	MdAE	41.2	47.4	43.2
	MPE	4.75	5.76	3.62
	MdPE	2.77	3.07	2.57

**Table 3 ijms-22-09194-t003:** The accuracy of retention index prediction for mid-polar stationary phases.

Model	Metric	Data Sets	
		DB-624	OV-17
This work	RMSE	36.8	43.8
	MAE	24.3	30.7
	MdAE	16.7	22.5
	MPE	2.33	2.52
	MdPE	1.66	1.73
Previous results [11,24]	RMSE	54.2	58.8
	MAE	32.5	47.3
	MdAE	19.2	42.9
	MPE	2.96	4.15
	MdPE	2.10	3.61

**Table 4 ijms-22-09194-t004:** The accuracy of retention index prediction for second dimension retention times ^a^.

Model	Metric	Data Sets	
		Test Set	External Set
This work	RMSE	0.229	0.211
	MAE	0.149	0.151
	MdAE	0.096	0.104
	MPE	4.34	4.17
	MdPE	2.89	3.07
Previous results [28]	RMSE	0.26	0.23
	MAE	0.17	0.16
	MPE	5	4

^a^—RMSE, MAE, MdAE are given in seconds.

**Table 5 ijms-22-09194-t005:** The accuracy of retention index prediction for SEDB624, DB-1701, and DB-210 data sets.

Metric	Data Sets		
	SEDB624	DB-1701	DB-210
RMSE	26.2	56.1	64.2
MAE	15.9	37.1	50.1
MdAE	9.95	20.0	42.3
MPE	2.03	2.54	5.05
MdPE	1.17	1.52	4.12

## Data Availability

The source code, pre-trained model parameters, source code of simple graphical user interface, and corresponding instructions are provided in the online repository: https://doi.org/10.6084/m9.figshare.14602317.

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
