# Peer review of "Deep Learning Based Prediction of Gas Chromatographic Retention Indices for a Wide Variety of Polar and Mid-Polar Liquid Stationary Phases"

_ijms, 2021, doi:10.3390/ijms22179194_

Round 1
Reviewer 1 Report
This manuscript could be accepted for publication in IJMS as well as in other journals focused on analytical chemistry and/or QSAR. The novelty of presented research (prediction of gas chromatographic retention indices based on compound structure is an important task for analytical chemistry) and it's impact for the scientific community are high. The introduction provide sufficient background and include all relevant references. The research methodology is adequate. The results are clearly presented. The conclusions supported by the data. The manuscript good illustrated and interesting to read. English language and style are fine. Very nice that the source code and pre-trained models are available online.
It would be good to add some additional metrics except RMSE, MAE, MdAE, MPE, and MdPE for validation of the suggested model.
I think that this manuscript will be of interest to the readers of the journal and I hope that the authors will continue their research in this direction.
Author Response
Reviewer 1
We appreciative of the reviewer for the high assessment of our work.
> It would be good to add some additional metrics except RMSE, MAE, MdAE, MPE, and MdPE for validation of the suggested model.
These metrics cover almost the full spectrum of metrics that can be used in this situation. However, we added R2 (coefficient of determination) and such error value so that the prediction error would be less than this value for 90% of the compounds. We added these values to sections 2.1 (polyethylene glycol stationary phases) and 2.2 (DB-624 stationary phase).
Reviewer 2 Report
The present work describes an application of deep learning in the modelling of retention behaviour in gas-chromatography. The paper is well written, and the proposed strategy provided a significant improvement in the prediction of the retention indices in gas-chromatographic columns with different polarity, compared to other approaches proposed previously.
However, before publication the authors should clarify the following points.
Lines 53-54. A brief description of deep learning should be given here.
Line 108, Table 1. The meaning of the acronyms reported in this Table (given in Paragraph 3.4) should be anticipated.
Lines 231-254. The authors took inspiration from the solvatochromic model (line 237) which implies a multilinear relationship between log k of a given solute and the features of the stationary phase. If I am not wrong, the retention index is not linearly related with log k. However, the authors obtained good multilinear models for two columns (DB-1071 and DB-210). Please, comment this point. A two-parameter model was developed for the first column, while a 5-parameter one was built for the second column. The authors should describe which was the criterion adopted to define the optimal complexity and give some statistical parameters to describe the quality of these models.
Being in a deep learning context, the authors should also consider the following artificial neural network approaches previously used to transfer retention data among different gas-chromatographic systems:
D'Archivio, A.A., Giannitto, A., Maggi, M.A. Cross-column prediction of gas-chromatographic retention of polybrominated diphenyl ethers (2013) Journal of Chromatography A, 1298, pp. 118-131.
D'Archivio, A.A., Incani, A., Ruggieri, F. Cross-column prediction of gas-chromatographic retention of polychlorinated biphenyls by artificial neural networks (2011) Journal of Chromatography A, 1218 (48), pp. 8679-8690.
